# Mineralogical Prediction on the Flotation Behavior of Copper and Molybdenum Minerals from Blended Cu–Mo Ores in Seawater

**Yoshiyuki Tanaka [1,2,*], Hajime Miki [2,*], Gde Pandhe Wisnu Suyantara [2] , Yuji Aoki [1] and Tsuyoshi Hirajima [3]**

[1]  Sumitomo Metal Mining Co., LTD., 17-5 Isouracho, Ehime, Niihama 792-0002, Japan; yuji.aoki.m3@smm-g.com

[2]  Department of Earth Resources Engineering, Kyushu University, 744 Motooka, Nishi-ku, Fukuoka 819-0395, Japan; pandhe@mine.kyushu-u.ac.jp

[3]  Sumitomo Metal Mining Co., LTD., 5-11-3 Shinbashi, Minatoku, Tokyo 105-8716, Japan; hirajima52@hotmail.co.jp

\*   Correspondence: yoshiyuki.tanaka.u8@smm-g.com (Y.T.); miki@mine.kyushu-u.ac.jp (H.M.); Tel.: +81-92-802-3349 (H.M.)

**Abstract:** The copper ore in Chilean copper porphyry deposits is often associated with molybdenum minerals. This copper–molybdenum (Cu–Mo) sulfide ore is generally mined from various locations in the mining site; thus, the mineral composition, oxidation degree, mineral particle size, and grade vary. Therefore, in the mining operation, it is common to blend the ores mined from various spots and then process them using flotation. In this study, the floatability of five types of Cu–Mo ores and the blending of these ores in seawater was investigated. The oxidation degree of these Cu–Mo ores was evaluated, and the correlation between flotation recovery and oxidation degree is presented. Furthermore, the flotation kinetics of each Cu–Mo ore were calculated based on a mineralogical analysis using mineral liberation analysis (MLA). A mineralogical prediction model was proposed to estimate the flotation behavior of blended Cu–Mo ore as a function of the flotation behavior of each Cu–Mo ore. The flotation results show that the recovery of copper and molybdenum decreased with the increasing copper oxidization degree. In addition, the recovery of blended ore can be predicted via the flotation rate equation, using the maximum recovery ($R_{max}$) and flotation rate coefficient ($k$) determined from the flotation rate analysis of each ore before blending. It was found that $R_{max}$ and $k$ of the respective minerals slightly decreased with increasing the degree of copper oxidation. Moreover, $R_{max}$ varied greatly depending on the mineral species. The total copper and molybdenum recovery were strongly affected by the degree of copper oxidation as the mineral fraction in the ore varied greatly depending upon the degree of oxidation.

**Keywords:** flotation; copper-molybdenum minerals; oxidation; seawater; mineralogical prediction

## 1. Introduction

Copper porphyry deposits are the most important and typical copper source for copper mines. It has been reported that more than 95% of copper mines in Chile are copper porphyry deposits [1]. Typical copper porphyry deposits comprise oxidized ore zones, secondary sulfide ore zones, and primary sulfide ore zones from the surface to the deep. Each zone contains characteristic copper minerals, e.g., chalcopyrite ($CuFeS_2$) and bornite ($Cu_5FeS_4$) in the primary sulfide ore zone, chalcocite ($Cu_2S$) and covellite (CuS) in the secondary sulfide ore zone, and atacamite ($Cu_2(OH)_3Cl$) and natural copper (Cu) in the oxide ore zone [1]. In addition, these copper minerals are often associated with molybdenite ($MoS_2$), which is the main molybdenum source, and both copper and molybdenum minerals are recovered [2–4].

The initial beneficiation stage of these copper and molybdenum minerals commonly includes flotation [3,5]. Flotation is widely used in mineral processing to separate minerals

based on the difference in surface hydrophobicity. In the conventional flotation circuit, the copper sulfide minerals are collected as a froth product. Meanwhile, the other hydrophilic and oxide minerals are separated as a sink. If required, the copper oxide minerals can be recovered as froth with sulfurization.

Generally, the copper porphyry ores are mined from several locations in the mining sites; thus, the mineral composition, oxidation degree, and copper grade vary greatly, which can affect the flotation performance. To overcome this problem, it is a common practice in a mining operation to blend these ores and provide a stable feed composition for maintaining an optimum flotation condition. However, the blending is carried out empirically, and the mixing ratio may change daily. Therefore, it is important to predict the effect of mixing ratio and mineralogical composition on the flotation rate and recovery, as well as the mineral grade, in advance. Allahkarami et al. [6] estimated the copper and molybdenum grade and recoveries in an industrial flotation plant using an artificial neural network (ANN). However, this ANN method did not consider the mineralogical aspect of the feed and the flotation behavior of individual copper and molybdenum minerals as the input parameter. On the other hand, Tijsseling et al. [7] used a mineralogical study to predict the flotation performance. However, their study focused on sediment-hosted copper–cobalt sulfide ore. Therefore, it is necessary to estimate the flotation behavior of blended ore on the basis of the mineralogical composition of each ore.

Conversely, seawater usage demand for mineral processing, including flotation processes, has recently increased, and various flotation estimation tests in seawater are necessary. Seawater or saline water has been used in the Las Luces copper–molybdenum (Cu–Mo) beneficiation plant in Taltal, Chile, in the Michilla Project, Chile, and the KCGM Project, Australia for processing sulfide minerals [8–11]. Although Alvarez and Castro [12] and Castro [13] showed that the use of seawater does not affect the flotation of pure chalcopyrite, various studies have shown that seawater contains various alkali metals ions that influence the flotation behavior of copper and molybdenum minerals [14–25]. Previous work by Hirajima et al. [26], Suyantara et al. [24], and Li et al. [27] showed that the colloidal magnesium hydroxide precipitate was the most detrimental ingredient for chalcopyrite and molybdenite flotation in seawater at high pH. However, there are limited studies available on the prediction of flotation performance of copper and molybdenum minerals from various Cu–Mo ores in seawater.

Fullston et al. [28] revealed that the degree of oxidation of copper-containing sulfide minerals can be ranked using zeta potential analysis in water, and they concluded that chalcocite is the most oxidized, followed by bornite, covellite, and chalcopyrite. This means that chalcocite and bornite are more easily oxidized than chalcopyrite. In addition, the various oxidation degrees of chalcopyrite, bornite, and molybdenite affect the flotation behavior of these minerals [29–31]. However, these studies were carried out using pure minerals; thus, it is important to study the effect of the degree of oxidation of copper and molybdenum minerals from various Cu–Mo ores.

In this study, the effect of the degree of oxidation on the recovery of copper and molybdenum minerals from various Cu–Mo ores in seawater was investigated. Furthermore, the effect of the mixing ratio of various Cu–Mo ores and mineralogical compositions on the flotation recovery was evaluated. It might be hypothesized that the flotation kinetics of blended Cu–Mo ore in seawater can be estimated by the flotation kinetics of each copper and molybdenum mineral from each Cu–Mo ore. Therefore, a mineralogical prediction model was proposed to estimate the flotation behavior of blended Cu–Mo ore on the basis of the flotation behavior of each Cu–Mo ore in this study.

## 2. Materials and Methods

### 2.1. Materials

The five ore samples used in the flotation tests were prepared and labeled as Samples A, B, C, D, and E. Samples A and B were from the primary sulfide zone, samples C and D were from the secondary sulfide zone, and Sample E was from the oxide zone in the copper

porphyry deposit in Chile. The ore samples were crushed using a jaw crusher (passing 6 mm) at the operating mine, crushed under 1.68 mm using a roll-crusher, and then packed in bags. Each bag contained a 0.875 kg ore sample, which was used for the flotation tests. Each ore sample was stored at $-40\,^\circ$C to minimize oxidation. The ore samples were thawed before being used for the flotation tests. The seawater for flotation was taken from Niihama, Ehime Prefecture, Japan, and the chemical composition of seawater is presented in Table 1.

**Table 1.** Chemical composition of seawater.

|  | Chemical Species | mg/L |  | Chemical Species | mg/L |
|---|---|---|---|---|---|
| 1 | Li | <1 | 15 | V | <1 |
| 2 | B | $4.5 \pm 0.14$ | 16 | Mn | <1 |
| 3 | C | <1000 | 17 | Fe | <1 |
| 4 | $NO_2^-$ | $0.42 \pm 0.013$ | 18 | Co | <1 |
| 5 | F | <10 | 19 | Ni | <1 |
| 6 | Na | $9800 \pm 294$ | 20 | Cu | <1 |
| 7 | Mg | $1400 \pm 42$ | 21 | Zn | <1 |
| 8 | Al | <1 | 22 | As | <1 |
| 9 | Si | <1 | 23 | Br | LOD |
| 10 | $PO_4$ | <10 | 24 | Sr | $7 \pm 0.2$ |
| 11 | $SO_4^{2-}$ | $898 \pm 27$ | 25 | Mo | <1 |
| 12 | Cl | $17,000 \pm 510$ | 26 | I | LOD |
| 13 | K | $400 \pm 12$ | 27 | Ba | <1 |
| 14 | Ca | $400 \pm 11$ |  |  |  |

LOD: limit of detection.

## 2.2. Analysis of Mineral Composition

Analysis of the mineral composition of the flotation feed and products was conducted using XRD (Malvern PANalytical, X'Pert Pro, EA Almelo, The Netherlands) and mineral liberation analysis (FEI, MLA650F, United States of America). A mineral list for MLA was prepared to identify the minerals on the basis of the minerals detected by XRD and the mineral database, which contains elemental compositions of approximately 500 minerals. The flotation tail contained low-grade copper and molybdenum and had a wide particle size distribution, which may affect the MLA results. Therefore, the flotation products were sieved using a 106 μm and 20 μm Tyler sieves to accurately analyze the mineral composition. The MLA measurements were conducted using two modes (i.e., extended back-scattered electron (XBSE) mode and sparse phase liberation (SPL) mode). The XBSE mode was used to analyze all minerals in the sample, and the SPL mode was used to analyze the mineral composition of copper and molybdenum. The samples were solidified with bakelite resin (sumilite resin@ PR-50252, Sumitomo Bakelite Co., Ltd., Tokyo, Japan) and then mixed with phenol resin (MultiFast Black, Struers Co., Ltd., Struer, Denmark) for hot solidification. The solidified fractions were polished with water-resistant abrasive papers (Struers Co., Ltd., Struer, Denmark) and SiC paper (Struers Co., Ltd., Struer, Denmark), before buffing with diamond suspensions (Struers Co., Ltd., Struer, Denmark) to a mirror finish, which were used as samples for MLA. The standard deviation for mineral liberation analysis was estimated at ca. 1.3%

## 2.3. Chemical Assay for Each Sample

Chemical analysis via inductively coupled plasma atomic emission spectroscopy (ICP-AES, Agilent ICP-AES 5100, Santa Clara, CA, USA) was conducted for all ore samples. The total copper (TCu) and total molybdenum (TMo) were analyzed by an alkaline melting method, in which samples were mixed with sodium peroxide ($Na_2O_2$, Kanto Chemical Co., Ltd., Tokyo, Japan) and melted at 800 °C. The sample obtained by alkali melting was dissolved in 10% dilute hydrochloric acid (HCl, Kanto Chemical Co., Inc., Tokyo, Japan) and analyzed by ICP-AES. The standard deviation for chemical analysis was estimated at ca. 0.03%.

Analysis of acid-soluble copper (Sol. Cu) was conducted by mixing 1 g of ore sample with 50 mL of 1 M citric acid, and the mixture was stirred at 130 rpm for 1 h. The entire volume was adjusted to 100 mL via the addition of water. The mixture was filtered using filter paper with a pore diameter of 1 µm. The copper grade in the recovered filtrate was measured using ICP-AES. The acid-soluble copper and molybdenum assay relates to the copper and molybdenum minerals associated with the oxide minerals [32]. Therefore, the oxidation degree of copper or molybdenum in the ore was estimated as the ratio of acid-soluble Cu or Mo to the total Cu or Mo (Equations (1) and (2)).

$$\text{Copper oxidation degree} = \frac{\text{Sol. Cu}}{\text{TCu}} \times 100. \tag{1}$$

$$\text{Molybdenum oxidation degree} = \frac{\text{Sol. Mo}}{\text{TMo}} \times 100. \tag{2}$$

### 2.4. Flotation Experiments

All flotation tests were conducted using a Denver-type flotation machine. A thionocarbamate (MX7017, Cytec Industries Inc., Woodland Park, NJ, USA) was used as the copper collector, diesel oil (diesel, Idemitsu Kosan Co., Ltd., Tokyo, Japan) was used as the molybdenum collector, methyl isobutyl carbinol (MIBC, Kanto Chemical Co., Ltd., Tokyo, Japan) was used as the frother, and slaked lime (Fujifilm Wako Pure Chemical Co., Ltd., Tokyo, Japan) was used as the pH modifier. The dosages of these reagents in the flotation tests were referenced to the dosage at the operating mine.

The ore samples (0.875 kg) were mixed with 500 mL of seawater in a stainless-steel (SUS304) mill with stainless-steel rods (SUS304). Pulp density was 57% (*w/w*). Thionocarbamate (30 g/t) and diesel oil (15 g/t) were added and ground for 10 min. Subsequently, the grinding medium was changed from the stainless-steel rod to the steel ball. The grinding time with the steel ball was adjusted for each ore (Table 2) to achieve a $P_{80}$ of 170 µm.

**Table 2.** Grinding time to produce $P_{80}$ of 170 µm for flotation feed.

| Sample | SUS Rod Min | Steel Ball Min | Total Min |
|---|---|---|---|
| A | 10 | 5.44 | 15.4 |
| B | 10 | 4.26 | 14.3 |
| C | 10 | 9.59 | 19.6 |
| D | 10 | 15.4 | 25.4 |
| E | 10 | 13.9 | 23.9 |

The ground product was fed into a 500 g flotation cell. Seawater was added to adjust the pulp density to 33 wt.%, and then MIBC (15 g/t) was added. The conditioning time was 1 min. Slaked lime was then added to adjust the pulp pH to 8.5. The flotation test was conducted for 30 min and the froth was collected every 3 min, 8 min, 15 min, and 30 min. Each product was filtered, dried at 110 °C, and weighed. Figure 1 shows the flowsheet of the flotation experiment.

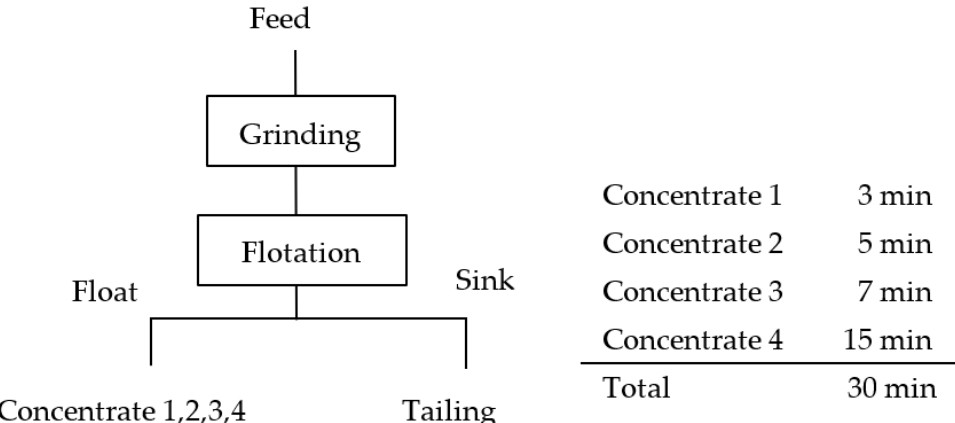

**Figure 1.** Flowsheet of flotation experiment procedure.

## 3. Results and Discussion

### 3.1. Characterization of Ore Samples

3.1.1. Chemical Assay of Each Ore Sample

Table 3 shows the chemical analysis by ore type. The copper grade ranged from 0.23–0.59%. The molybdenum grade ranged from 0.01–0.22%. Sample B had the highest Mo grade compared to the other ore samples. The degree of Cu oxidation (Sol. Cu/TCu) was as high as 38.7% for sample E from the oxide zone, followed by samples D and C from the secondary sulfide zone at 15.6% and 3.5%, respectively. Samples A and B, sampled from the primary sulfide zone, where the effect of oxidation was the least, had the lowest oxidation degree of 1.7% and 1.5%, respectively. Table 3 shows that there was no significant difference in the degree of molybdenum oxidation (Sol. Mo/TMo) for all ore samples.

**Table 3.** Chemical assays for each ore sample (%).

| Sample | A | B | C | D | E |
|---|---|---|---|---|---|
| Total Cu (%) | $0.47 \pm 0.01$ | $0.59 \pm 0.02$ | $0.23 \pm 0.01$ | $0.32 \pm 0.01$ | $0.31 \pm 0.01$ |
| Sol Cu (%) | $0.008 \pm 0.000$ | $0.009 \pm 0.000$ | $0.008 \pm 0.000$ | $0.050 \pm 0.002$ | $0.12 \pm 0.004$ |
| $\frac{\text{Sol Cu}}{\text{Total Cu}} \times 100(\%)$ | 1.70 | 1.50 | 3.50 | 15.60 | 38.70 |
| Total Mo (%) | $0.011 \pm 0.000$ | $0.220 \pm 0.01$ | $0.071 \pm 0.000$ | $0.010 \pm 0.000$ | $0.013 \pm 0.000$ |
| Sol Mo (%) | <0.001 | <0.001 | 0.003 | <0.001 | 0.001 |
| $\frac{\text{Sol Mo}}{\text{Total Mo}} \times 100(\%)$ | - | - | 4.20 | - | 7.70 |

3.1.2. Mineral Composition

Table 4 lists the major minerals in each flotation feed ore identified by XRD analysis, and Table 5 presents the mineral composition obtained by the XBSE mode of MLA. MLA analysis results showed that feldspars were the most abundant in all ore samples, with a maximum of 50% (sample D) and a minimum of 35% (sample C). The presence of feldspar was confirmed by the XRD analysis results (Table 4). In addition, Table 5 demonstrates that quartz was the second most abundant mineral and was present in the range of 18–21%. MLA analysis revealed the presence of chalcopyrite in Samples A, B, C, and D. However, there was no chalcopyrite detected in Sample E, likely due to the low concentration of chalcopyrite in the sample. Therefore, MLA was performed using the sparse phase liberation analysis mode to identify copper and molybdenum minerals in the ore samples.

**Table 4.** Mineralogical analysis result of each ore sample measured by XRD.

| Minerals | | Sample | | | | |
|---|---|---|---|---|---|---|
| | | **A** | **B** | **C** | **D** | **E** |
| Feldspar group | Albite | ◎ | nd | ◎ | ◎ | ◎ |
| | Orthoclase | nd | nd | nd | nd | nd |
| | Microcline | × | ○ | × | ◎ | ◎ |
| Mica group | Biotite | nd | nd | nd | nd | nd |
| | Muscovite | nd | nd | nd | nd | nd |
| | Phlogopite | ◎ | nd | ○ | ○ | ○ |
| Chlorite group | Montmorillonite | nd | nd | nd | nd | nd |
| | Almandine | nd | nd | nd | nd | nd |
| | Clinochlore | ○ | ○ | ◎ | ◎ | × |
| Other minerals | Quartz | ◎ | ◎ | ◎ | ◎ | ◎ |
| | Clinoferrosilite | × | ◎ | × | × | × |
| | Pyrite | ○ | ◎ | ◎ | ◎ | ◎ |
| | Magnetite | ◎ | ◎ | × | ○ | ◎ |
| | Anhydrite | nd | nd | nd | nd | nd |

◎: strong detection, ○: medium detection, ×: weak detection, nd: no detection.

**Table 5.** Mineral composition for each ore sample (wt.%).

| Minerals | Sample A | Sample B | Sample C | Sample D | Sample E |
|---|---|---|---|---|---|
| Feldspar | 43.10 ± 0.55 | 46.90 ± 0.60 | 35.43 ± 0.45 | 49.98 ± 0.63 | 42.41 ± 0.54 |
| Quartz | 17.89 ± 0.23 | 20.15 ± 0.26 | 21.44 ± 0.27 | 18.68 ± 0.24 | 18.88 ± 0.24 |
| Biotite | 17.71 ± 0.22 | 0.69 ± 0.01 | 10.9 ± 0.14 | 15.79 ± 0.20 | 16.60 ± 0.21 |
| Pyrite | 1.94 ± 0.02 | 10.89 ± 0.14 | 9.84 ± 0.12 | 0.69 ± 0.01 | 0.89 ± 0.01 |
| Fe oxide/hydroxide | 8.91 ± 0.11 | 2.27 ± 0.03 | 8.25 ± 0.10 | 7.24 ± 0.09 | 14.17 ± 0.18 |
| Chlorite | 1.92 ± 0.02 | 1.70 ± 0.02 | 3.58 ± 0.05 | 2.12 ± 0.03 | 1.74 ± 0.02 |
| Muscovite | 2.00 ± 0.03 | 1.01 ± 0.01 | 3.36 ± 0.04 | - | 0.86 ± 0.01 |
| Other silicate | 1.48 ± 0.02 | 1.01 ± 0.01 | 1.79 ± 0.02 | 1.17 ± 0.01 | 1.62 ± 0.02 |
| Clay | - | - | 1.67 ± 0.02 | - | - |
| Chalcopyrite | 3.28 ± 0.04 | 5.59 ± 0.07 | 1.22 ± 0.02 | 1.04 ± 0.01 | - |
| Molybdenite | - | 0.77 ± 0.01 | - | - | - |
| Clinochlore | - | 8.24 ± 0.10 | 0.64 ± 0.01 | - | 0.58 ± 0.01 |
| Calcite | - | - | 0.50 ± 0.01 | - | - |
| Gypsum | - | - | - | 0.57 ± 0.01 | - |
| Others | 1.77 ± 0.02 | 1.78 ± 0.02 | 1.38 ± 0.03 | 2.72 ± 0.03 | 2.26 ± 0.03 |
| Total | 100 | 100 | 100 | 100 | 100 |

Table 6 presents the MLA analysis using the sparse phase liberation analysis mode. The MLA analysis demonstrated that the ore samples contained chalcopyrite, bornite, covellite, chalcocite, atacamite, and native copper. The molybdenum minerals in the ore samples were molybdenite and molybdenum oxide ($MoO_3$). Table 6 demonstrates that samples A and B contained 92.2% and 99.2% of copper as chalcopyrite, respectively. Chalcopyrite is a characteristic of primary sulfide ores. Sample C, from the secondary sulfide zone, had a lower percentage of chalcopyrite (90.0%) compared to samples A and B. In sample C, 4.4% and 2.2% of copper were identified as chalcocite and covellite, which are characteristic minerals of the secondary sulfide ore zone. Sample D (i.e., from the secondary sulfide ore zone) contained 53.9% copper, which was identified as chalcopyrite. Sample D contained 13.1% and 3.2% chalcocite and covellite, respectively, which were higher than those of sample C. In addition, Sample D contained 5.0% atacamite, which is generally present in the oxide ore zone. Therefore, the ore type of Sample D was closer to the oxide ore zone than that of Sample C. Chalcopyrite composition decreased to 24.9% and chalcocite composition increased to 29.4% in Sample E, which was collected from the oxide ore zone. The atacamite and native copper compositions were 23.8% and 5.6%, respectively.

The presence of atacamite and native copper in Sample E indicates that sample E mostly consisted of minerals from the oxide zone.

**Table 6.** Copper and molybdenum mineral composition for each ore sample (wt.%).

| | Sample | A | B | C | D | E |
|---|---|---|---|---|---|---|
| | chalcopyrite | 92.20 ± 1.17 | 99.20 ± 1.26 | 90.00 ± 1.14 | 53.90 ± 0.68 | 24.90 ± 0.32 |
| | bornite | 4.00 ± 0.05 | 0.33 ± 0.00 | 3.38 ± 0.04 | 24.6 ± 0.31 | 8.96 ± 0.11 |
| Cu | covellite | 1.97 ± 0.03 | 0.31 ± 0.00 | 2.19 ± 0.03 | 3.16 ± 0.04 | 7.37 ± 0.09 |
| minerals | chalcocite | 1.86 ± 0.02 | 0.13 ± 0.00 | 4.43 ± 0.06 | 13.10 ± 0.17 | 29.40 ± 0.37 |
| | atacamite | 0.00 ± 0.00 | 0.01 ± 0.00 | 0.00 ± 0.00 | 4.97 ± 0.06 | 23.80 ± 0.30 |
| | native copper | 0.02 ± 0.00 | 0.06 ± 0.00 | 0.00 ± 0.00 | 0.23 ± 0.00 | 5.60 ± 0.07 |
| | Total | 100 | 100 | 100 | 100 | 100 |
| Mo | molybdenite | 96.70 ± 1.23 | 99.80 ± 1.27 | 96.00 ± 1.22 | 99.90 ± 1.27 | 75.20 ± 0.96 |
| minerals | Mo oxide | 3.30 ± 0.04 | 0.20 ± 0.00 | 4.00 ± 0.05 | 0.10 ± 0.00 | 24.80 ± 0.31 |
| | Total | 100 | 100 | 100 | 100 | 100 |

More than 96% of the molybdenum minerals in Samples A, B, C, and D were molybdenite and the rest were molybdenum oxide. The molybdenite composition in Sample E was as low as 75.2%, and the molybdenum oxide composition was 24.8%. This result indicates that the molybdenum in Sample E was more oxidized than that in the other ore samples.

From the degree of Cu oxidation (Sol. Cu/TCu) presented in Table 3 and the mineral composition listed in Table 6, the relationship between oxidation degree and mineral ratio was calculated, as shown in Figure 2. Copper sulfide minerals were mostly present as chalcopyrite if the oxidation degree was low. As the oxidation degree increased, the chalcopyrite composition decreased, the composition of chalcocite, bornite, and atacamite gradually increased. Furthermore, when the oxidation degree was 15% or more, the oxidation of secondary sulfide minerals increased and the presence of native copper increased. According to these results, the ore samples were oxidized in the following order of samples: B > A > C > D > E. It is possible that primary copper sulfide, chalcopyrite, gradually changes to bornite and chalcocite with increasing oxidation degree. Under stronger oxidizing conditions, chalcopyrite and bornite decompose to chalcocite, covellite, atacamite, and then native copper.

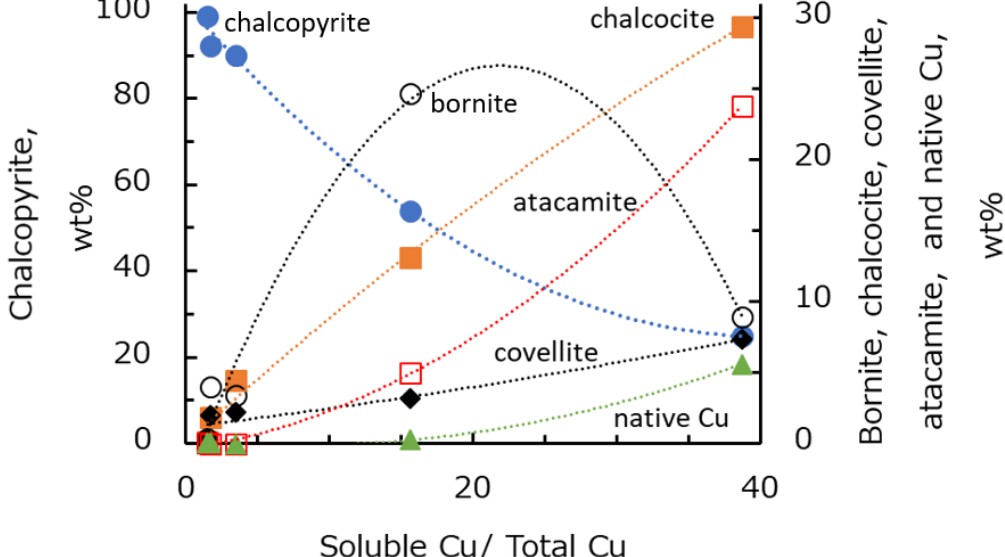

**Figure 2.** Relationships between copper oxidation degree and composition of copper minerals.

*3.2. Flotation*

3.2.1. Flotation Recovery

Figure 3 shows the relationship between the flotation time and recovery of copper and molybdenum by ore type. The copper recovery of Samples A, B, C, D, and E increased significantly during the first 10 min of flotation time (Figure 3a). Under this condition, Sample A had the highest copper recovery, followed by Samples B, C, D, and E. However, after 30 min of flotation, sample B had a higher final recovery compared to sample A, followed by samples C, D, and E. The reason for the higher final recovery of Sample B was likely the higher copper grade of Sample B (Table 3) compared to that of other ore samples. Interestingly, the flotation trend of copper from each ore followed the trend of copper mineral composition presented in Table 6. By comparing Table 6 and Figure 3, it can be seen that the copper recovery decreased with decreasing proportion of chalcopyrite as the main copper sulfide mineral in each ore. Additionally, a significant loss of copper recovery in Sample D and E was correlated with an increasing proportion of copper oxide minerals. Indeed, the relationship between the copper oxidation degree and copper recovery presented in Figure 4a clearly shows that there was a negative correlation between the recovery of copper and the degree of copper oxidation, i.e., the copper recovery decreased with an increase in the degree of copper oxidation.

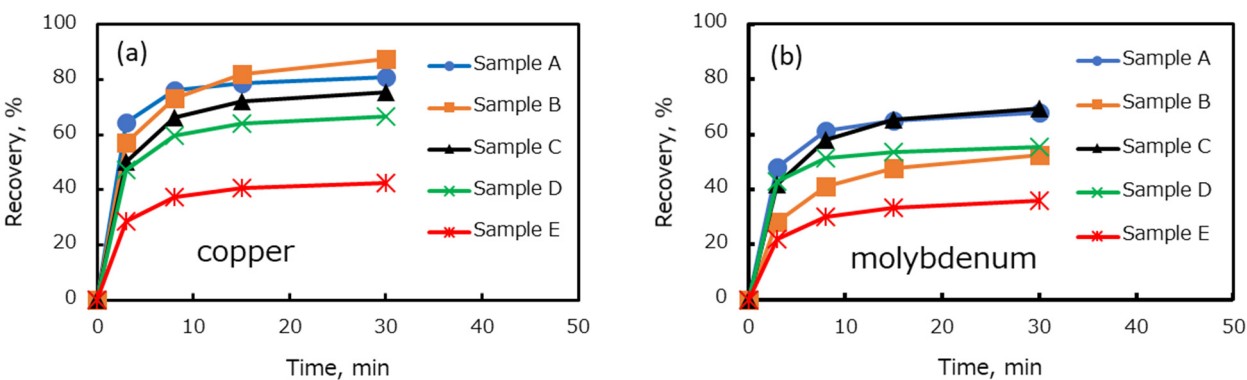

**Figure 3.** Flotation recovery of copper (**a**) and molybdenum (**b**) for each ore sample.

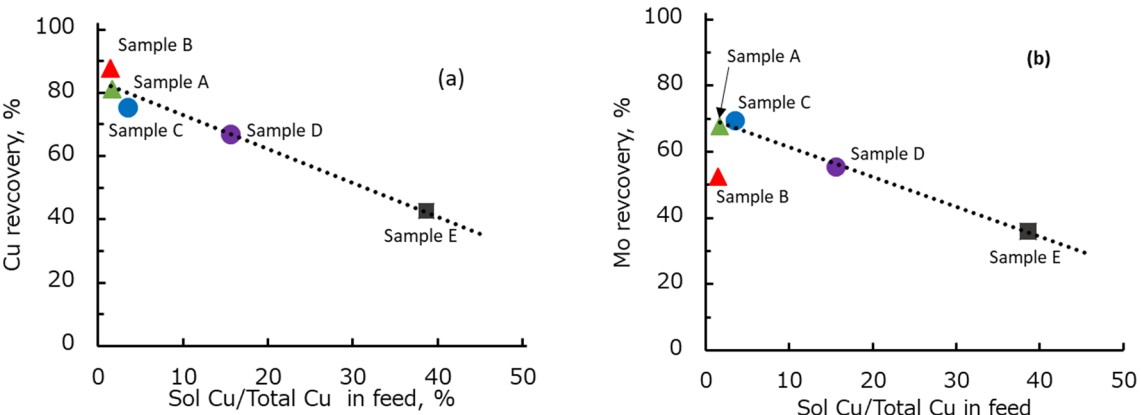

**Figure 4.** Relationships between copper oxidation degree and the recovery of copper (**a**) and molybdenum (**b**) in each ore.

Figure 3b shows that the recovery of molybdenum followed a similar trend to the recovery of copper, except for Sample B. After 30 min of flotation, Samples A and C exhibited a relatively similar recovery and the molybdenum recovery gradually decreased in Samples D, B, and E. One possible answer for this result is the oxidation of molybdenum minerals. However, total soluble Mo as one of the indications of molybdenum oxidation (Table 3) shows an insignificant change of soluble Mo in all ore samples. Moreover, the

molybdenum mineral composition (Table 6) shows that the molybdenum oxide in each ore was less than 5% and only 25% in Sample E. Therefore, it is unlikely that the decrease in molybdenum recovery was caused mainly by the mineral oxidation and molybdenum mineral composition. A correlation between copper oxidation degree and molybdenum recovery was made to understand the molybdenum recovery trend, and the result is presented in Figure 4b. Figure 4b demonstrates that there was a negative correlation between the copper oxidation degree and molybdenum recovery, i.e., the molybdenum recovery decreased with an increase in the degree of copper oxidation. This negative correlation indicates that copper ions from oxidized copper minerals might suppress the floatability of molybdenum. It is well known that oxidized copper has a naturally acidic pH. Therefore, these copper ions might be dissolved during the grinding process of the oxidized Cu–Mo ore at low pH and then precipitated on the surface of molybdenum minerals as the pH increased in the flotation tests.

Figure 4b shows that the recovery of sample B was outside the linear trend owing to low molybdenum recovery. The low molybdenum recovery of Sample B can be attributed to the higher molybdenum grade in Sample B (i.e., 0.22%) compared to that of other ore types (Table 3). However, the dosage of diesel oil was the same for all ore samples used in this study. Therefore, it is supposed that the dosage of diesel oil to float the molybdenum minerals in Sample B, which had a higher molybdenum grade than other ore samples, was insufficient. To prove this argument, a flotation test was conducted with a higher diesel oil dosage. The diesel oil dosage was increased from 15 to 126 g/t in the flotation of Sample B. The molybdenum recovery improved from 52% to 59%.

### 3.2.2. Flotation Kinetics and Recovery of Copper and Molybdenum

According to the flotation results shown in Figure 3, the maximum recovery ($R_{max}$) and flotation kinetics coefficient ($k$) were calculated using the model formula shown in Equation (3). This equation is a classical first-order model of flotation kinetics [32–35]. $R_{max}$ indicates the maximum recovery that can be obtained for a particular mineral and flotation system, whereas $k$ indicates the flotation rate of a mineral. This $k$ value can be used to predict how fast a mineral can reach the maximum recovery. The detailed derivation of this equation can be seen in Dowling et al. [33]. $R_{max}$ and $k$ were derived using the "solver" option in the Microsoft EXCEL program, i.e., by minimizing the sum of the squared error between the calculated recovery and the actual recovery obtained in the flotation.

$$R = R_{max}\left(1 - e^{-kt}\right). \tag{3}$$

Table 7 shows the calculation results for $R_{max}$ and $k$ by ore type. The $R_{max}$ value of copper decreased in the order of Sample B and Sample A from the primary sulfide zone, followed by Samples C and D from the secondary sulfide zone, and Sample E from the oxidized zone. The $R_{max}$ value of Mo decreased in the following order of samples: C > A > D > B > E. The $k$ values varied in the range of 0.369–0.557 for copper and 0.245–0.523 for molybdenum.

**Table 7.** Maximum recovery ($R_{max}$) and flotation kinetics coefficient ($k$) of copper and molybdenum in each ore sample.

| Sample | Maximum Recovery, $R_{max}$ | | | | | Kinetics, $k$ | | | | |
|---|---|---|---|---|---|---|---|---|---|---|
| | A | B | C | D | E | A | B | C | D | E |
| Cu | 0.791 | 0.833 | 0.730 | 0.645 | 0.409 | 0.557 | 0.355 | 0.369 | 0.422 | 0.384 |
| Mo | 0.656 | 0.503 | 0.668 | 0.539 | 0.344 | 0.426 | 0.245 | 0.304 | 0.523 | 0.316 |

### 3.2.3. Floatability of Minerals

Sample E was used to understand the floatability of each mineral in each ore type. Sample E was chosen because the mineral composition of Sample E can be representative

of all minerals present in all ore types (Table 6). Figure 5 shows the relationship between the flotation time and recovery of chalcopyrite, bornite, chalcocite, covellite, atacamite, and native copper in Sample E. Figure 5 shows that the recovered minerals were chalcopyrite, chalcocite, covellite, and bornite. Meanwhile, atacamite and native copper were the least floatable minerals. This result was caused by the usage of thionocarbamate as a collector. The thionocarbamate is actively adsorbed on copper sulfide minerals compared to copper oxide minerals, thus improving the recovery of copper sulfide minerals.

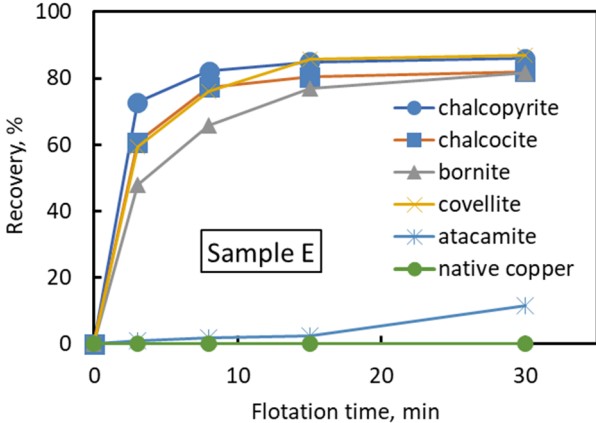

**Figure 5.** Flotation recovery for each copper mineral (Sample E).

Table 8 shows the calculation results of $R_{max}$ and $k$ for each mineral obtained by applying Equation (3). In addition, the $R_{max}$ and $k$ values of each mineral in the other ore samples are listed in this table. Table 8 shows that the $R_{max}$ values of chalcocite and covellite were higher than those of other copper minerals, followed by bornite and chalcopyrite. This result indicates that chalcocite and covellite are more floatable compared to bornite and chalcopyrite. This result might be caused by the absence of iron in chalcocite and covellite. The presence of various iron oxidation products in chalcopyrite and bornite has been reported to reduce the floatability of chalcopyrite and bornite [29,36,37]. The $R_{max}$ values of native copper and atacamite were low because native copper and atacamite are hydrophilic oxide minerals and, thus, less floatable.

**Table 8.** Maximum recovery, $R_{max}$, and kinetics, $k$, for each mineral with flotation in each ore sample.

| | Maximum Recovery, $R_{max}$ | | | | | Kinetics, $k$ | | | | |
|---|---|---|---|---|---|---|---|---|---|---|
| | **A** | **B** | **C** | **D** | **E** | **A** | **B** | **C** | **D** | **E** |
| Chalcopyrite | 0.763 | 0.835 | 0.723 | 0.670 | 0.704 | 0.536 | 0.351 | 0.376 | 0.360 | 0.395 |
| Chalcocite | 0.908 | 0.905 | 0.859 | 0.833 | 0.808 | 0.490 | 0.140 | 0.409 | 0.652 | 0.462 |
| Covellite | 0.908 | 0.944 | 0.574 | 0.843 | 0.850 | 0.699 | 0.363 | 0.154 | 0.384 | 0.376 |
| Bornite | 0.811 | 0.833 | 0.693 | 0.673 | 0.786 | 0.459 | 0.868 | 0.299 | 0.403 | 0.282 |
| Atacamite | – | 0.050 | – | 0.018 | 0.050 | – | 0.008 | – | 0.069 | 0.008 |
| Native Cu | 0.000 | 0.000 | – | 0.087 | 0.000 | 0.000 | 0.000 | – | 0.022 | 0.000 |
| Molybdenite | 0.885 | 0.503 | 0.684 | 0.570 | 0.411 | 0.649 | 0.246 | 0.304 | 0.491 | 0.323 |
| Mo oxide | 0.909 | 0.163 | 0.150 | 0.369 | 0.043 | 0.228 | 0.176 | 0.519 | 1.033 | 0.037 |

"–" indicates that the amount of mineral was too low to record.

Table 8 demonstrates that the $R_{max}$ of each mineral slightly decreased with the increasing oxidation degree of the ore sample. For instance, the $R_{max}$ of chalcopyrite in Sample B, the least oxidized ore sample, was the highest (i.e., 0.835). The $R_{max}$ of chalcopyrite was decreased in the more oxidized ore samples (Sample D and E). A similar trend was observed for other minerals. The slightly decreasing trend of $R_{max}$ for each mineral might have been caused by the increasing proportion of hydrophilic minerals (i.e., atacamite and

native copper) in the ore. The effect of molybdenum oxidation degree on $R_{max}$ and $k$ of molybdenum minerals exhibited no clear trend.

### 3.2.4. Flotation Performance of Blended Ore Sample

According to the production plan in each operating mine, the flotation feed is generally a blended ore from various spots of copper porphyry deposits in the mining site. In this study, the recovery of the blended ore sample was estimated using the recovery of each ore sample. Sample F was prepared by blending the five ore types (Samples A, B, C, D, and E) with a mixture ratio, as presented in Table 9. The mixture ratio was chosen on the basis of the mixture ratio used in the flotation plant. Flotation tests were conducted, and the $R_{max}$ and $k$ values were determined.

**Table 9.** Blending ratio of sample F (%).

| Sample | A | B | C | D | E | F |
|---|---|---|---|---|---|---|
| Blending ratio | 34.4 | 19.2 | 21.7 | 16.4 | 8.3 | 100 |

The flotation results are shown in Figure 6. The actual copper and molybdenum recoveries in this figure were obtained from the flotation recovery of sample F. The estimated copper and molybdenum recoveries were obtained by weighted averaging of $R_{max}$ and $k$ for each ore type, as presented in Table 7. Equation (4) was used to calculate the estimated recovery of blended ore. The estimated $R_{max}$ ($R_{max, estimated}$) and $k$ ($k_{estimated}$) values were calculated using Equations (5) and (6), respectively. The weighting factor (w) is the blending ratio of the ore samples, and index $i$ denotes each ore sample.

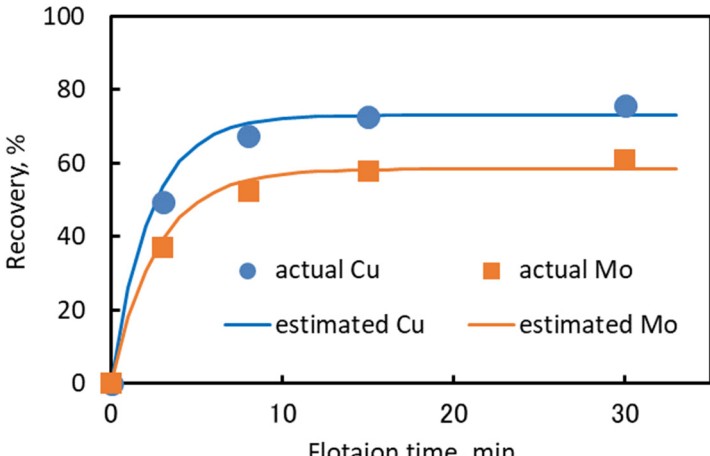

**Figure 6.** Comparison of actual and estimated flotation recovery of sample F for copper and molybdenum.

A comparison of the actual and estimated values of $R_{max}$ and $k$ is presented in Table 10. Table 10 shows that the estimated $R_{max}$ and $k$ values were relatively similar to the actual $R_{max}$ and $k$ values. The recovery profile presented in Figure 6 shows that the estimated recovery of copper and molybdenum obtained from the estimated values of $R_{max}$ and $k$ values could fit the actual recovery of copper and molybdenum. This result suggests that the floatability of a blended ore can be estimated from the floatability of each ore type by applying the blending ratio as a weighting factor of $R_{max}$ and $k$.

$$R_{estimated} = R_{max,\ estimated}\left(1 - e^{-k_{estimated}t}\right). \tag{4}$$

$$R_{max,\ estimated} = \sum_i w_i R_{max,\ i}. \tag{5}$$

$$k_{estimated} = \sum_i w_i k_{\,i}. \tag{6}$$

**Table 10.** Comparison of actual and estimated value on maximum recovery ($R_{max}$) and flotation kinetics coefficient ($k$).

| | Maximum Recovery ($R_{max}$) | | Flotation Kinetics Coefficient ($k$) | |
|---|---|---|---|---|
| | **Actual** | **Estimated** | **Actual** | **Estimated** |
| Cu | 0.737 | 0.730 | 0.359 | 0.441 |
| Mo | 0.591 | 0.584 | 0.311 | 0.372 |

A similar method was applied to each mineral in the blended ore sample (sample F). The recovery of each copper and molybdenum mineral in the blended ore sample was estimated using the flotation parameters ($R_{max}$ and $k$) of each ore type (Equation (7)). The estimated $R_{max}$ and $k$ of each copper and molybdenum mineral (j) in each ore type (i) was calculated using Equations (8) and (9). $g_{f,i}$ is the copper or molybdenum feed grade of ore $i$ as presented in Table 3. $x_{i,j}$ is the composition of mineral $j$ in ore $i$ (Table 6). $R_{max,\text{ij}}$ and $k_{\text{ij}}$ are the $R_{max}$ and $k$ values of mineral $j$ in ore $i$ as presented in Table 8, and $w_i$ is the blending ratio of ore $i$ as shown in Table 9. Table 11 compares the estimated $R_{max}$ and $k$ of various minerals in the blended ore with the actual values. The estimated $R_{max}$ and $k$ values of each mineral in the blended ore calculated using Equations (7)–(9) were relatively close to the actual $R_{max}$ and $k$ values of each mineral.

**Table 11.** Comparison of actual (Sample F) and estimated value on maximum recovery ($R_{max}$) and flotation kinetics coefficient ($k$) for each mineral.

| | Maximum Recovery ($R_{max}$) | | Flotation Kinetics Coefficient ($k$) | |
|---|---|---|---|---|
| **Sample** | **Actual** | **Estimated** | **Actual** | **Estimated** |
| Chalcopyrite | 0.767 | 0.772 | 0.355 | 0.437 |
| Chalcocite | 0.733 | 0.838 | 0.568 | 0.524 |
| Covellite | 0.825 | 0.838 | 0.370 | 0.473 |
| Bornite | 0.690 | 0.725 | 0.294 | 0.407 |
| Atacamite | 0.010 | 0.040 | 0.060 | 0.026 |
| Native Cu | 0.100 | 0.006 | 0.022 | 0.064 |
| Molybdenite | 0.596 | 0.568 | 0.310 | 0.596 |
| Mo oxide | 0.248 | 0.212 | 0.283 | 0.248 |

Figure 7a,b shows the flotation results of copper and molybdenum minerals in Sample F and the calculated results (solid line) as a function of the estimated values in Table 11. Figure 7 demonstrates that the actual recovery of each mineral in the blended ore could be fitted with the estimated recovery of each mineral from each ore type. This result suggests that the recovery of each mineral in each ore type can be used to predict the recovery of each mineral in the blended ore.

$$R_{estimated,\ j} = R_{max,\ estimated,\ j}\left(1 - e^{-k_{estimated,\ j}t}\right). \tag{7}$$

$$R_{max,\ estimated,\ j} = \frac{\sum_i g_{f,i} x_{ij} w_i R_{max,\ ij}}{\sum_i g_{f,i} x_{ij} w_i}. \tag{8}$$

$$k_{estimated,j} = \frac{\sum_i g_{f,i} x_{ij} w_i k_{\,ij}}{\sum_i g_{f,i} x_{ij} w_i}. \tag{9}$$

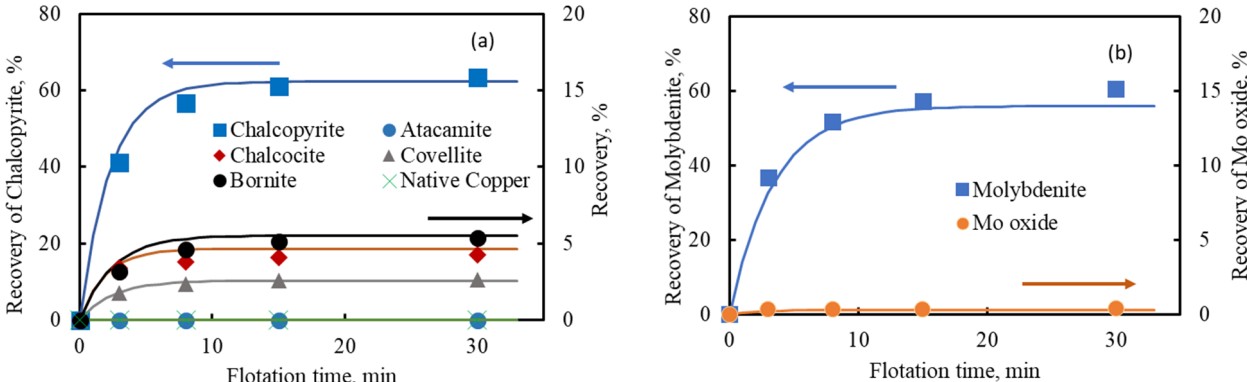

**Figure 7.** Comparison of actual and estimated flotation recovery of each copper mineral (**a**) and each molybdenum mineral (**b**) in Sample F.

## 4. Conclusions

In this study, five types of Cu–Mo ore with different degrees of oxidation were collected from operated copper–molybdenum mines. The floatability test of individual and blended ores was carried out in seawater. Mineralogical analysis was performed to identify the copper and molybdenum minerals in each Cu–Mo ore. It was shown that the recovery of copper and molybdenum in seawater decreased with increasing copper oxidation degree in each Cu–Mo ore. In addition, the dissolved copper from copper oxidation during the grinding at low pH might precipitate on the surface of copper and molybdenum minerals at pH 8.5, thus suppressing the floatability of copper and molybdenum minerals. Mineralogical analysis showed that the copper and molybdenum minerals in each Cu–Mo ore were chalcopyrite, bornite, chalcocite, covellite, atacamite, native copper, molybdenite, and molybdenum oxide. The proportion of atacamite and native copper in the Cu–Mo ore was strongly correlated with the degree of copper oxidation.

The mineralogical prediction was performed on the basis of the flotation kinetics parameters (i.e., kinetics coefficient ($k$) and maximum recovery ($R_{max}$)) of each copper and molybdenum mineral in the various Cu–Mo ores. The mineralogical prediction was used to estimate the flotation kinetics parameters of blended Cu–Mo ore. It was shown that a simple mineralogical prediction based on the $R_{max}$ and $k$ of each copper and molybdenum minerals in the various Cu–Mo ores, the blending ratio of each Cu–Mo ore, total Cu or Mo and the mineral composition of the feed ore can be used to estimate the flotation recovery of blended Cu–Mo ore. This estimation method might be useful for industrial applications to predict the flotation behavior of blended Cu–Mo ore if the proportion of each Cu–Mo ore is known.

**Author Contributions:** Conceptualization, Y.T., T.H., and H.M.; methodology and experiments, Y.T. and Y.A.; writing—original draft preparation, Y.T.; writing—review and editing, T.H., G.P.W.S., and H.M.; supervision, T.H. and H.M.; project administration, Y.T. All authors have read and agreed to the published version of the manuscript.

**Funding:** This research received no external funding.

**Institutional Review Board Statement:** Not applicable.

**Informed Consent Statement:** Not applicable.

**Data Availability Statement:** Not applicable.

**Acknowledgments:** We appreciate the support of this research by Sumitomo Metal Mining Co., Ltd., for providing samples and useful advice. We also thank Yoshihisa Takahashi and Eri Takida for their cooperation in this study.

**Conflicts of Interest:** The authors declare no conflict of interest.

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
