# Peer review of "Mineralogical Prediction on the Flotation Behavior of Copper and Molybdenum Minerals from Blended Cu–Mo Ores in Seawater"

_minerals, doi:10.3390/min11080869_

Round 1
Reviewer 1 Report
- At Abstract, ore blending is important factor? then the reason should be added.
Instead of blending and kinetics , it is better to show the purpose of flotation, recovery rate, the effect of sea water and whether ore types are oxide or sulfide,
- From Abstract and introduction, I can’t find the purpose of this paper
- At introduction, I think that only sulfide minerals could be floated. Explain what kind of minerals are recovered by flotation. There is no principle on flotation.
- At lines 72-73 in page 2, what is the reason that various ores are mixed?
- At line 88 in page 2, did these ores come from Chile? Add the country name of mine to your paper
- At lines 228-239 in page 7, chemical formulae of various mineral ores should be expressed at the beginning of this paper
- At lines 278-279 in page 8, you explained the floatability of molybdenum was affected by copper ions but I think it depends on the oxidation of molybdenum. Check it again.
- From equation(3), you need to show the graph of logR and time. And you need to explain how you introduce this equation. From the kinetic coefficient and maximum recovery, what do you want to explain?
- From Table 8, the same minerals Rmax at each ores are different. Explain the reason.
- At lines 385-387 in page 11, explain the reason that you select the mixture ratio like Table 9.
- At lines 419-422 in page 12, the total recovery of copper and molybdenum minerals was 100% → the matching rate between the actual value and the estimated value on the total recovery of copper and molybdenum minerals was 100%
- Conclusion should be rewritten out. It is too long. I think that the key point is the predictability of the flotation behavior of a blended ore by using your estimation method. For this, you had a flotation experiments and got the actual results. By using this result, you introduce the kinetic equation and got the estimated value at each minerals. And you applied this result to a blended ore and check whether both values coincided with high accuracy. With these key points, you can rewrite your title, abstract, introduction and conclusion better.
Author Response
Dear Reviewer 1, thanks for your precious comments. due to your comments, paper became more comprehensive. we improved our manuscripts and we made reply from your comments as below, and manuscript are attached. Corrected part is indicated as red color.
Reviewer 1
- At Abstract, ore blending is important factor? then the reason should be added. Instead of blending and kinetics , it is better to show the purpose of flotation, recovery rate, the effect of sea water and whether ore types are oxide or sulfide,
Authors’ comment: Thank you for your comment. We revised the abstract in the original manuscript to improve the clarity and connections between abstract and the main manuscript.
“Copper sulphide ore is generally mined from various locations in the mining site; thus, the mineral composition, oxidation degree, mineral particle size, and grade vary. Therefore, in the mining operation, it is common to blend the ores mined from various spots and then process them using flotation. In this study, floatability of five types of copper and molybdenum (Cu-Mo) ores and blending of these ores in seawater were investigated. The oxidation degree of these Cu-Mo ores was evaluated and the correlation between flotation recovery and oxidation degree is presented. Furthermore, the flotation kinetics of each Cu-Mo ores was calculated based on the mineralogical analysis using mineral liberation analysis (MLA). A mineralogical prediction model was proposed to estimate the flotation behaviour of blended Cu-Mo ore based on the flotation behaviour of each Cu-Mo ore. The flotation results show that the recovery of copper and molybdenum decreased with the increasing copper oxidization degree. In addition, the recovery of blended ore can be predicted via the flotation rate equation, using the maximum recovery (Rmax) and flotation rate coefficient (k) determined from the flotation rate analysis of each ore before blending. It was found that Rmax and k of the respective minerals did not significantly change with the degree of copper oxidation. Moreover, Rmax varied greatly depending on the mineral species. The recovery was strongly affected by the degree of copper oxidation as the mineral fraction in the ore varied greatly depending upon the degree of oxidation.”
- From Abstract and introduction, I can’t find the purpose of this paper
Authors’ comment: We added this information in revised manuscript. Please see the Introduction.
“In this study, the effect of the degree of oxidation on the recovery of copper and molybdenum minerals from various Cu-Mo ores in seawater was investigated. Besides, the effect of the mixing ratio of various Cu-Mo ores and mineralogical compositions on the flotation recovery was evaluated. It might be hypothesized that the flotation kinetics of blended Cu-Mo ore in seawater can be estimated by the flotation kinetics of each copper and molybdenum mineral from each Cu-Mo ore. Therefore, a mineralogical prediction model was proposed to estimate the flotation behaviour of blended Cu-Mo ore based on the flotation behaviour of each Cu-Mo ore in this study. ”
- At introduction, I think that only sulfide minerals could be floated. Explain what kind of minerals are recovered by flotation. There is no principle on flotation.
Authors’ comment: Yes, you are correct. However, the oxide minerals can be floated by first altering their surface hydrophilicity. We added this information in the revised manuscript.
“The initial beneficiation stage of these copper and molybdenum minerals is commonly using flotation [3,5]. Flotation is widely used in mineral processing to separate minerals based on the difference in surface hydrophobicity. In the conventional flotation circuit, the copper sulphide minerals are collected as a froth product. Meanwhile, the other hydrophilic and oxide minerals are separated as a sink. If required, the copper oxide minerals can be recovered as froth with sulfurization.”
- At lines 72-73 in page 2, what is the reason that various ores are mixed?
Authors’ comment: Generally, the copper porphyry ores are mined from several locations in the mining sites, thus, the mineral composition, oxidation degree, and copper grade vary greatly, which can affect the flotation performance. To overcome this problem, it is a common practice in a mining operation to blend these copper-molybdenum ores and provide a stable feed composition for maintaining an optimum flotation condition. We added this information in the Introduction of the revised manuscript.
“Generally, the copper porphyry ores are mined from several locations in the mining sites, thus, the mineral composition, oxidation degree, and copper grade vary greatly, which can affect the flotation performance. To overcome this problem, it is a common practice in a mining operation to blend these ores and provide a stable feed composition for maintaining an optimum flotation condition.”
- At line 88 in page 2, did these ores come from Chile? Add the country name of mine to your paper
Authors’ comment: Yes, we have added country name.
“Samples A and B were from the primary sulphide zone, samples C and D were from the secondary sulphide zone, and Sample E was from the oxide zone in the copper porphyry deposit in Chile.
- At lines 228-239 in page 7, chemical formulae of various mineral ores should be expressed at the beginning of this paper
Authors’ comment: Thank you, we added this information in the Introduction.
“Each zone contains characteristic copper minerals e.g. chalcopyrite (CuFeS2) and bornite (Cu5FeS4) in the primary sulphide ore zone, chalcocite (Cu2S) and covellite (CuS) in the secondary sulphide ore zone, and atacamite (Cu2(OH)3Cl) and natural copper (Cu) in the oxide ore zone [1]. In addition, these copper minerals are often associated with molybdenite (MoS2), which is the main molybdenum source and both copper and molybdenum minerals are recovered [2–4].”
- At lines 278-279 in page 8, you explained the floatability of molybdenum was affected by copper ions but I think it depends on the oxidation of molybdenum. Check it again.
Authors’ comment: Thank you for your question. We revise this section for clarify this issue.
“Figure 3b shows that the recovery of molybdenum followed a similar trend to the recovery of copper, except for Sample B. After 30 min of flotation, Samples A and C exhibited a relatively similar recovery and the molybdenum recovery gradually decreased in Samples D, B, and E. One possible answer for this result is the oxidation of molybdenum minerals. However, total soluble Mo as one of the indications of molybdenum oxidation (Table 3) shows an insignificant change of soluble Mo in all ore samples. Moreover, the molybdenum mineral composition (Table 6) shows that the molybdenum oxide in each ore was less than 5% and only 25% in Sample E. Therefore, it is unlikely the decrease of molybdenum recovery is caused mainly by the mineral oxidation and molybdenum mineral composition. A correlation between copper oxidation degree and molybdenum recovery was made to understand the molybdenum recovery trend and the result is presented in Figure 4b. Figure 4b demonstrates that there is a negative correlation between the copper oxidation degree and molybdenum recovery, i.e., the molybdenum recovery decreased with an increase in the degree of copper oxidation. This negative correlation indicates that copper ions from oxidized copper minerals might suppress the floatability of molybdenum.”
- From equation (3), you need to show the graph of logR and time. And you need to explain how you introduce this equation. From the kinetic coefficient and maximum recovery, what do you want to explain?
Authors’ comment: Equation (3) is a classical first-order model of flotation kinetics. The derivation of this equation has been explained in detail by Dowling et al. [3]. Because it is a well-known equation in flotation, therefore we think we don’t have to introduce this equation in more detail in the manuscript.
As it is not common to present the log R vs time for the flotation results, therefore, we present the fitting results in the format recovery vs time. Furthermore, this information can be interpreted straight forward to assess the flotation kinetics and maximum recovery. The Rmax indicates the maximum recovery that can be obtained for a particular mineral and flotation system. The kinetics coefficient indicates how fast a mineral reaches the maximum recovery and how long the flotation should be carried out to get the maximum recovery. We added this information in the revised manuscript.
“Based on the flotation results shown in Figure 3, the maximum recovery (Rmax) and flotation kinetics coefficient (k) were calculated using the model formula shown in Equation (3). This equation is a classical first-order model of flotation kinetics [32–35]. The Rmax indicates the maximum recovery that can be obtained for a particular mineral and flotation system. The k indicates the flotation rate of a mineral. This k value can be used to predict how fast a mineral can reach the maximum recovery. The detailed derivation of this equation can be seen in Dowling et al. [33]. The Rmax and k were derived by using the “solver” option in the Microsoft EXCEL program, i.e., by minimising the sum of the squared error between the calculated recovery and the actual recovery obtained in the flotation.”
- From Table 8, the same minerals Rmax at each ores are different. Explain the reason.
Authors’ comment: The Rmax parameter is affected by many factors e.g. the type of mineral, mineral composition, particle size distribution, oxidation degree, mineral liberation, and flotation system (flotation reagent and type flotation machine). It was shown that each ore has a different oxidation degree that might affect the Rmax value as we discussed in the manuscript.
“Table 8 demonstrates that the Rmax of each mineral slightly decreased with increasing the oxidation degree of the ore sample. For instance, the Rmax of chalcopyrite in Sample B, the least oxidized ore sample, was the highest (i.e., 0.835). The Rmax of chalcopyrite was decreased in the more oxidized ore samples (Sample D and E). A similar trend was observed for other minerals. The slightly decreasing trend of Rmax for each mineral might be caused by the increasing proportion of hydrophilic minerals (i.e., atacamite and native copper) in the ore. The effect of molybdenum oxidation degree on the Rmax and k of molybdenum minerals exhibited no clear trend.”
- At lines 385-387 in page 11, explain the reason that you select the mixture ratio like Table 9.
Authors’ comment: Thank you for your question. The mixture ratio was chosen based on the mixture ratio in the real flotation plant. We added this information in the revised manuscript.
“Sample F was prepared by blending the five ore types (Samples A, B, C, D, and E) with a mixture ratio, as presented in Table 9. The mixture ratio was chosen based on the mixture ratio used in the flotation plant.”
- At lines 419-422 in page 12, the total recovery of copper and molybdenum minerals was 100% → the matching rate between the actual value and the estimated value on the total recovery of copper and molybdenum minerals was 100%
Authors’ comment: Thank you for raising this issue. We deleted this part to avoid confusion.
- Conclusion should be rewritten out. It is too long. I think that the key point is the predictability of the flotation behavior of a blended ore by using your estimation method. For this, you had a flotation experiments and got the actual results. By using this result, you introduce the kinetic equation and got the estimated value at each minerals. And you applied this result to a blended ore and check whether both values coincided with high accuracy. With these key points, you can rewrite your title, abstract, introduction and conclusion better.
Authors’ comment: Thank you very much for your suggestions. We have revised the conclusion as below:
“In this study, five types of Cu-Mo ore with different degrees of oxidation were collected from operated copper-molybdenum mines. The floatability test of individual and blended ores was carried out in seawater. Mineralogical analysis was performed to identify the copper and molybdenum minerals in each Cu-Mo ore. It was shown that the recovery of copper and molybdenum in seawater decreased with increasing copper oxidation degree in each Cu-Mo ore. In addition, the dissolved copper from copper oxidation might suppress the floatability of copper and molybdenum minerals. Mineralogical analysis shows that the copper and molybdenum minerals in each Cu-Mo ore were chalcopyrite, bornite, chalcocite, covellite, atacamite, native copper, molybdenite, and molybdenum oxide. The proportion of atacamite and native copper in the Cu-Mo ore is strongly correlated with the degree of copper oxidation.
The mineralogical prediction was performed based on the flotation kinetics parameters (i.e., kinetics coefficient (k) and maximum recovery (Rmax)) of each copper and molybdenum mineral in the various Cu-Mo ores. The mineralogical prediction was used to estimate the flotation kinetics parameters of blended Cu-Mo ore. It was shown that a simple mineralogical prediction based on the Rmax and k of each copper and molybdenum minerals in the various Cu-Mo ores and the blending ratio of each Cu-Mo ore can be used to estimate the flotation recovery of blended Cu-Mo ore. This estimation method might be useful for industrial applications to predict the flotation behaviour of blended Cu-Mo ore if the proportion of each Cu-Mo ore is known.”
Reviewer 2 Report
for the discussion section from curiosity: how do the flotation rates you determined compare to classical ones for ores from the same mines? can you think of a way how to reach more than 80%? Especially in front of resource scarcity discussions and the need for copper this is a crucial question? please comment
can you give an outlook of water consumption of processing copper ore on global level?
paper has a very relevant topic, clear structure, good description of methods, sincere discussion, sound conclusions. can you formulate a research hypothesis, which you can verify or falsify in the conclusions?
Author Response
Dear Reviewer 2, thanks for your positive comments. Your comments encourage us and paper became more comprehensive. we improved our manuscripts and we made reply from your comments as below, and manuscript are attached. Corrected part is indicated as red color.
Reviewer 2
- for the discussion section from curiosity: how do the flotation rates you determined compare to classical ones for ores from the same mines? can you think of a way how to reach more than 80%? Especially in front of resource scarcity discussions and the need for copper this is a crucial question? please comment
Authors’ comment: Thanks for your comment. We use a classical first-order model of flotation kinetics in this study. Therefore, we cannot make any comparison as requested by the reviewer.
Recovering copper from a mining operation depends on the type of copper mineral and mineral composition. For a copper-molybdenum ore with mainly primary copper sulphide or secondary copper sulphide, most of the copper sulphide minerals can be recovered by the addition of other copper collectors or increased the collector’s dosage. On the other hand, if the copper-molybdenum ore contains oxidized copper minerals, these minerals require further treatment to improve their surface hydrophobicity e.g. sulfurization using NaHS. We agree that the improvement of copper recovery is a very important topic as raised by the reviewer’s comment. However, this work is mainly focused on studying the flotation kinetics of various copper minerals from various copper-molybdenum ore s, therefore, we cannot discuss how to improve the copper recovery in the manuscript.
- can you give an outlook of water consumption of processing copper ore on global level?
Authors’ comment: Thank you for your question.We don’t have any actual data on water consumption for processing copper-molybdenum ore at the global level. However, based on a survey in 2000, the water consumption for processing valuable minerals in Chile reached 823,000 m3/day. As the ore grade is decreasing, we expect that this number will increase more significantly which will affect the supply of fresh water in the future. To mitigate this problem, and also due to the scarcity of fresh water in desert places, at which many mining operations are located in Chile, Peru, Australia and, Western USA, seawater has been used for processing the copper-molybdenum ore.
- paper has a very relevant topic, clear structure, good description of methods, sincere discussion, sound conclusions. can you formulate a research hypothesis, which you can verify or falsify in the conclusions?
Authors’ comment: Thank you for your comment. The hypothesis is that the flotation kinetics of blended copper-molybdenum ore in seawater can be estimated by the flotation kinetics of each copper and molybdenum mineral from each copper-molybdenum ore. Table 11 shows that the actual value of Rmax and k of blended copper-molybdenum ore is close enough to the estimated value of Rmax and k. Therefore, the hypothesis is verified.
We added the hypothesis in the Introduction of the revised manuscript.
“It might be hypothesized that the flotation kinetics of blended Cu-Mo ore in seawater can be estimated by the flotation kinetics of each copper and molybdenum mineral from each Cu-Mo ore”
Reviewer 3 Report
The authors have tried to analyse the flotation process for copper-molybdenum ore with different oxidation levels using seawater. There are a few points mentioned below that need to be addressed before publication:
- There are several English errors in the manuscript. Please fix all language errors in the manuscript.
- Surface characterization and zeta potential are the crucial characterizations that influence the process. The author must include it in the study.
- Please highlight the novelty aspect of the present research in the introduction, abstract and conclusion sections.
- Abstract: Please try to avoid abbreviations in the abstract and conclusion.
- Introduction: The author should mention the current problem in the flotation of copper-molybdenum ore in seawater and must link to the recent research.
- It is not clear why the authors have carried out few flotation tests on kinetics only?
- There are several other parameters that influence the separation need to be listed in the experimental section.
- What is the error range in the measurement with analysis? Please mention with error %.
- Please include the details of the analytical setup along with the error involved in measurement.
- MLA data needs to explain the interlocking pattern as well as the deportment of Cu and Mo.
- Flotation parameters: On what basis authors have selected these parameters to keep constant for the flotation experimentation?
- Why have the authors not chosen any statistically designed experimentation? It is better to do full factorial or orthogonal design in such cases.
- Size analysis and Size-wise assay/mineral concentration of the feed samples must be given and should correlate with the separated products.
- The results explained in Table and Figures must be given along with standard deviation/error.
- Most of the places in the discussion is mostly explained the values rather than the science/mechanism. Please improve the discussion on that front.
- The English language needs to be given attention and request authors to refine sentences with the right choice of words. The manuscript needs a thorough revision of its language and style. Overall, this paper is challenging to read. Avoid redundancies and keep it short. I suggest a comprehensive overhaul of the text for a more precise understanding of the reader.
- References: There are many articles on this subject. The author needs to review all of these published articles on Cu-Mo flotation with emphasis on the present research subject. Please revisit the cited references carefully and request to include all the relevant ones.
Author Response
Dear Reviewer 3, thanks for your precious comments. due to your comments, paper became more comprehensive. we improved our manuscripts and we made reply from your comments as below, and manuscript are attached. Corrected part is indicated as red color.
Reviewer 3
The authors have tried to analyse the flotation process for copper-molybdenum ore with different oxidation levels using seawater. There are a few points mentioned below that need to be addressed before publication:
- There are several English errors in the manuscript. Please fix all language errors in the manuscript.
Authors’ comment: Thank you for your comment. We improve the English and minimize the mistake including grammatical errors.
- Surface characterization and zeta potential are the crucial characterizations that influence the process. The author must include it in the study.
Authors’ comment: Thank you for your opinion, we agree that surface characterization and zeta potential are crucial characterizations. However, this surface characterization and zeta potential measurement might be suitable for pure minerals. For a mixture of various minerals, such as the copper-molybdenum ores presented in this study, the surface characterization and zeta potential information would be difficult to interpret. Besides, it is difficult to measure the zeta potential in the seawater due to it has a very high ionic strength. Therefore, we focus on the evaluation of flotation kinetics of copper and molybdenum minerals from various copper-molybdenum ores.
- Please highlight the novelty aspect of the present research in the introduction, abstract and conclusion sections.
Authors’ comment: We revised the abstract, introduction, and conclusion to highlight the novelty. Please see the revised manuscript.
- Abstract: Please try to avoid abbreviations in the abstract and conclusion.
Authors’ comment: Thank you for your suggestion, we corrected MLA and XRD as Mineral Liberation Analysis (MLA) and X-ray Diffraction (XRD) in the revised manuscript.
- Introduction: The author should mention the current problem in the flotation of copper-molybdenum ore in seawater and must link to the recent research.
Authors’ comment: Thank you for your suggestion. We address this in the revised manuscript as follow:
“Conversely, seawater usage demand for mineral processing, including flotation processes, has recently increased and various flotation estimation tests in seawater are necessary. Seawater or saline water has been used in the Las Luces copper-molybdenum (Cu-Mo) beneficiation plant in Taltal, Chile, in the Michilla Project, Chile, and the KCGM Project, Australia for processing sulphide minerals [8–11]. Although Alvarez and Castro [12], Castro [13] showed that the use of seawater does not affect the flotation of pure chalcopyrite, various research showed that seawater contains various alkali metals ions that influence the flotation behaviour of copper and molybdenum minerals [14–25]. The previous work, Hirajima et al. [26], Suyantara et al. [24], and Li et al. [27] showed that the colloidal magnesium hydroxide precipitate was the most detrimental ingredient for chalcopyrite and molybdenite flotation in seawater at high pH. However, there is a limited study available on the prediction of flotation performance of copper and molybdenum minerals from various Cu-Mo ores in seawater. ”
- It is not clear why the authors have carried out few flotation tests on kinetics only?
Authors’ comment: The flotation kinetics was calculated based on the individual copper and molybdenum minerals. We used the mineral liberation analysis (MLA) to collect the mineralogical data of each fraction of the flotation product. Each fraction was sieved using a 106 and 20 Tyler sieve to accurately analyze the mineral composition. These samples were then prepared for MLA analysis (e.g., mixing with resin, briquetting, and polishing). The MLA results were then used to calculate the recovery of each mineral in each copper-molybdenum ores. As we used five types of copper-molybdenum ores, the analytical work would be enormous and time-consuming. Therefore, we can just provide a few flotation tests on the kinetics study.
- There are several other parameters that influence the separation need to be listed in the experimental section.
Authors’ comment: Thank you for this important suggestion. We agree that many other parameters might influence the separation (e.g., flotation reagents and dosage, air flowrate, particle size distribution, mineral composition, etc.). However, the purpose of this research is to investigate flotation behavior on various real ore and its blending under a real processing condition. Therefore, we tried to minimize the influence of other parameters by using a fixed parameter for all flotation tests.
- What is the error range in the measurement with analysis? Please include the details of the analytical setup along with the error involved in measurement.
Authors’ comment: We already put the details of the analytical setup in the original manuscript. For the error involved in the measurement, we added the following information in the revised manuscript.
“The standard deviation for mineral liberation analysis was estimated at ca. 1.3%”
“The standard deviation for chemical analysis was estimated at ca. 0.03%.”
- MLA data needs to explain the interlocking pattern as well as the deportment of Cu and Mo.
Authors’ comment: Thanks for your suggestion. We agree that this interlocking and the deportment of Cu and Mo might affect the flotation. However, we used this data for another manuscript that is under review. Therefore, we focused on using MLA to provide the composition of copper and molybdenum minerals in each copper-molybdenum ores in this study.
- Flotation parameters: On what basis authors have selected these parameters to keep constant for the flotation experimentation?
Authors’ comment: This study was performed to support the study of the flotation behavior of copper-molybdenum ore used in the real flotation plant. Therefore, we were using the flotation parameters that represent the real flotation condition.
- Why have the authors not chosen any statistically designed experimentation? It is better to do full factorial or orthogonal design in such cases.
Authors’ comment: Thank you for the interesting question. The objective of this study is to provide a simple method based on mineralogical analysis to predict the flotation behavior of blended copper-molybdenum ore in seawater and to compare the effect of oxidation degree of each copper-molybdenum ore. The flotation parameter and the blending ratio of copper-molybdenum ore were dictated by the condition of the real flotation plant, thus, the only variable to be investigated is the effect of mineral composition of each copper-molybdenum ores. Besides, there is a limitation to performed MLA analysis as the basic data to estimate the flotation behavior of blended copper-molybdenum ore. Therefore, we performed this study without any statistically designed experimentation such as full factorial design or orthogonal design. Despite this limitation, this study shows that a simple prediction method based on mineralogical analysis could estimate the real flotation behavior of blended copper-molybdenum ore.
- Size analysis and Size-wise assay/mineral concentration of the feed samples must be given and should correlate with the separated products.
Authors’ comment: Thank you for your suggestion. However, we used this information for another manuscript that is under reviewed. Therefore, we could not provide this information in this manuscript. We already discussed the particle size and Cu and Mo grade of feed sample (Table 3) in the manuscript. We hope this information is sufficient for your consideration.
“. The grinding time with the steel ball was adjusted for each ore (Table 2) to achieve a P80 of 170 µm.”
- The results explained in Table and Figures must be given along with standard deviation/error.
Authors’ comment: Thank you for your suggestion. As we explained in answer to question No. 6, due to some limitations for accessing mineral liberation analysis as the basis for evaluation of flotation behavior, we cannot provide a standard deviation/ error bar for all Tables and Figures. However, we added standard deviation values for Tables 1, 3, and 5 in the revised manuscript.
- Most of the places in the discussion is mostly explained the values rather than the science/mechanism. Please improve the discussion on that front.
Authors’ comment: we revised the discussion as suggested to improve the quality of the manuscript. Thank you for your suggestion.
- The English language needs to be given attention and request authors to refine sentences with the right choice of words. The manuscript needs a thorough revision of its language and style. Overall, this paper is challenging to read. Avoid redundancies and keep it short. I suggest a comprehensive overhaul of the text for a more precise understanding of the reader.
Authors’ comment: Thank you for your suggestion. We revised the manuscript and asked for a professional English edit service to improve the English quality. We hope this and our responses are sufficient for publication in Minerals.
- References: There are many articles on this subject. The author needs to review all of these published articles on Cu-Mo flotation with emphasis on the present research subject. Please revisit the cited references carefully and request to include all the relevant ones.
Authors’ comment: thank you for addressing this. We revisited the cited references and revised the manuscript to include the relevant references only.
Reviewer 4 Report
The authors of the paper undertook to describe the ore flotation process with variable flotation properties. This is a difficult task because the flotation process, at the same time, is influenced by many factors of a different nature.
In this manuscrypt, the introduction needs to be redrafted. The information presented in paper are too general. They should primarily concern the analysis of the literature for a similar ore and the ways of describing the flotation process, so that the reader can understand the results of the Authors' research.
There are many studies in the world literature about mineral flotation in saline waters, also with the use of sea water. It is necessary to include such content in this manuscript - please complete this information.
14 items of literature are not enough, especially since they are mainly self-citations. The number of cited manuscripts describing similar studies should be increased
I recommend the authors also to read the selected papers e.g.:
1. International Journal of Mineral Processing, Volume 148, 10 March 2016, Pages 48-58
2. Transactions of Nonferrous Metals Society of China 27 (10): 2260-2271. DOI: 10.1016 / S1003-6326 (17) 60252-8
3. Transactions of Nonferrous Metals Society of China
Volume 27, Issue 10, October 2017, Pages 2260-2271
3. Physicochemical Problems of Mineral Processing (2013) 49 (1), 341–356
4. Physicochemical Problems of Mineral Processing (2007) 41, 51-65
5. and papers also published in MDPI
Comments and questions
- either use SI units or explain the units used (such as "mesh")
- chapter titles cannot be identical - correction is required
-why low pulp density was used, "the pulp density to 33 wt%"
-line 192-193-please edit this sentence
- is the dashed line, in Fig. 4 an automatic line trend line or is it a model curve?
- what information did the authors obtain about this flotation process based on the value of "k"?
- what rules did the authors follow when preparing the mixed samples?
- to demonstrate the usefulness of the Rmax estimation methodology, it would be necessary to indicate / enter, define / indicator - please propose such a quantitative indicator
- the whole "conclusion" is an analysis of research results, unfortunately they are not conclusions! This section of the manuscript should be re-edited
Positive:
- research prepared and carried out very well, congratulations
- although there are many similar studies, this manuscript shows a different point of view on the flotation process of this ore,
- the proposed methodology for assessing the flotability of polymetallic and multi-mineral ore with a known mineralogical rock is relatively simple to apply, which is an advantage
Kind Regards,
Author Response
Dear Reviewer 4, thanks for your precious and positive comments. due to your comments, paper became more comprehensive. we improved our manuscripts and we made reply from your comments as below, and manuscript are attached. Corrected part is indicated as red color.
Reviewer 4
The authors of the paper undertook to describe the ore flotation process with variable flotation properties. This is a difficult task because the flotation process, at the same time, is influenced by many factors of a different nature.
- In this manuscrypt, the introduction needs to be redrafted. The information presented in paper are too general. They should primarily concern the analysis of the literature for a similar ore and the ways of describing the flotation process, so that the reader can understand the results of the Authors' research.
Authors’ comment: Thank you for your comment. We revised the introduction to improve the manuscript clarity and story flow. Please see the revised Introduction.
- There are many studies in the world literature about mineral flotation in saline waters, also with the use of sea water. It is necessary to include such content in this manuscript - please complete this information.
Authors’ comment: We agree with you, we added more references regarding the seawater in the revised manuscript.
“Conversely, seawater usage demand for mineral processing, including flotation processes, has recently increased and various flotation estimation tests in seawater are necessary. Seawater or saline water has been used in the Las Luces copper-molybdenum (Cu-Mo) beneficiation plant in Taltal, Chile, in the Michilla Project, Chile, and the KCGM Project, Australia for processing sulphide minerals [8–11]. Although Alvarez and Castro [12], Castro [13] showed that the use of seawater does not affect the flotation of pure chalcopyrite, various research showed that seawater contains various alkali metals ions that influence the flotation behaviour of copper and molybdenum minerals [14–25]. The previous work, Hirajima et al. [26], Suyantara et al. [24], and Li et al. [27] showed that the colloidal magnesium hydroxide precipitate was the most detrimental ingredient for chalcopyrite and molybdenite flotation in seawater at high pH. However, there is a limited study available on the prediction of flotation performance of copper and molybdenum minerals from various Cu-Mo ores in seawater. ”
- 14 items of literature are not enough, especially since they are mainly self-citations. The number of cited manuscripts describing similar studies should be increased
I recommend the authors also to read the selected papers e.g.:
1. International Journal of Mineral Processing, Volume 148, 10 March 2016, Pages 48-58
2. Transactions of Nonferrous Metals Society of China 27 (10): 2260-2271. DOI: 10.1016 / S1003-6326 (17) 60252-8
3. Transactions of Nonferrous Metals Society of China
Volume 27, Issue 10, October 2017, Pages 2260-2271
3. Physicochemical Problems of Mineral Processing (2013) 49 (1), 341–356
4. Physicochemical Problems of Mineral Processing (2007) 41, 51-65
5. and papers also published in MDPI
Chalcopyrite and Molybdenite Flotation in Seawater: The Use of Inorganic Dispersants to Reduce the Depressing Effects of Micas Minerals 2021, 11(5), 539; https://doi.org/10.3390/min11050539
Flotation Separation of Chalcopyrite and Molybdenite Assisted by Microencapsulation Using Ferrous and Phosphate Ions: Part II. Flotation Metals 2021, 11(3), 439; https://doi.org/10.3390/met11030439
Effects of Potassium Propyl Xanthate Collector and Sodium Sulfite Depressant on the Floatability of Chalcopyrite in Seawater and KCl Solutions Minerals 2020, 10(11), 991; https://doi.org/10.3390/min10110991
The Depressing Effect of Kaolinite on Molybdenite Flotation in Seawater Minerals 2020, 10(6), 578; https://doi.org/10.3390/min10060578
Authors’ comment: Thank you for your suggestion. We added more relevant references in the revised manuscript.
Comments and questions
- Either use SI units or explain the units used (such as "mesh")
Authors’ comment: Thank you, we corrected as mentioned, ex. 10 mesh-> 1.68 mm
- Chapter titles cannot be identical - correction is required
Authors’ comment: We checked and found that the title of section 2.3 and 2.4 are identical. We deleted one of these titles. Thank you very much for your suggestion.
- why low pulp density was used, "the pulp density to 33 wt%"
Authors’ comment: The pulp density in the rougher flotation of the real plant is 33%, therefore we use this pulp density in this study.
- line 192-193-please edit this sentence
Authors’ comment: Thank you for pointing this out. We revised this sentence, “that quartz was the second most abundant quartz” to “that quartz was the second most abundant mineral”.
- is the dashed line, in Fig. 4 an automatic line trend line or is it a model curve?
Authors’ comment: This is a trend line to improve the clarity, it is not from a mathematical model
- what information did the authors obtain about this flotation process based on the value of "k"?
Authors’ comment: k is the flotation kinetics coefficient, showing how fast the particular mineral could reach the Rmax. This information is important to determine the flotation time in the real plant.
- what rules did the authors follow when preparing the mixed samples?
Authors’ comment: Thank you for your question. The mixture ratio was chosen based on the mixture ratio in the real flotation plant. We added this information in the revised manuscript.
“Sample F was prepared by blending the five ore types (Samples A, B, C, D, and E) with a mixture ratio, as presented in Table 9. The mixture ratio was chosen based on the mixture ratio used in the flotation plant.”
- to demonstrate the usefulness of the Rmax estimation methodology, it would be necessary to indicate / enter, define / indicator - please propose such a quantitative indicator
Authors’ comment: The Rmax indicates the maximum recovery that can be obtained for a particular mineral and flotation system. Therefore, the estimated Rmax can be used to estimate how much a particular mineral can be collected as a product, to estimate the Rmax of blended ore, and what kind of strategy for further recovery improvement can be developed. Therefore, it is difficult to propose a quantitative indicator to assess the usefulness of Rmax.
- the whole "conclusion" is an analysis of research results, unfortunately they are not conclusions! This section of the manuscript should be re-edited
Authors’ comment: Thank you for your comment. We revised the conclusion.
“In this study, five types of Cu-Mo ore with different degrees of oxidation were collected from operated copper-molybdenum mines. The floatability test of individual and blended ores was carried out in seawater. Mineralogical analysis was performed to identify the copper and molybdenum minerals in each Cu-Mo ore. It was shown that the recovery of copper and molybdenum in seawater decreased with increasing copper oxidation degree in each Cu-Mo ore. In addition, the dissolved copper from copper oxidation might suppress the floatability of copper and molybdenum minerals. Mineralogical analysis shows that the copper and molybdenum minerals in each Cu-Mo ore were chalcopyrite, bornite, chalcocite, covellite, atacamite, native copper, molybdenite, and molybdenum oxide. The proportion of atacamite and native copper in the Cu-Mo ore is strongly correlated with the degree of copper oxidation.
The mineralogical prediction was performed based on the flotation kinetics parameters (i.e., kinetics coefficient (k) and maximum recovery (Rmax)) of each copper and molybdenum mineral in the various Cu-Mo ores. The mineralogical prediction was used to estimate the flotation kinetics parameters of blended Cu-Mo ore. It was shown that a simple mineralogical prediction based on the Rmax and k of each copper and molybdenum minerals in the various Cu-Mo ores and the blending ratio of each Cu-Mo ore can be used to estimate the flotation recovery of blended Cu-Mo ore. This estimation method might be useful for industrial applications to predict the flotation behaviour of blended Cu-Mo ore if the proportion of each Cu-Mo ore is known.”
Positive:
- research prepared and carried out very well, congratulations
- although there are many similar studies, this manuscript shows a different point of view on the flotation process of this ore,
- the proposed methodology for assessing the flotability of polymetallic and multi-mineral ore with a known mineralogical rock is relatively simple to apply, which is an advantage
Authors’ comment: Thanks for your positive comments. The manuscript became much better with your valuable insight.
Round 2
Reviewer 1 Report
- To emphasize blending ores, I suggest the title is “Mineralogical prediction on the flotation behavior of copper and molybdenum minerals from blended Cu-Mo ores in seawater”.
- At Abstract, there are no explanations on molybdenum ore and behavior. Add the reason why you use copper and molybdenum ores.
- At abstract, “Rmax and k of the respective minerals did not significantly change with the degree of copper oxidation” and “The recovery was strongly affected by the degree of copper oxidation” do not seem consistent with each other. You need to check it again with Table. 8.
- At lines 136-142 in page 4, you need to show the references for calculation of oxidation degree.
- At lines 293-296 in page 9, you explained the floatability of molybdenum was affected by copper ions. I think you need to explain the reason.
- There are no explanation on the effect of seawater. If you have any data on the flotation in tab water or distilled water, you may show the comparison results between two solutions.
- At conclusion of lines 481-482, I don’t think that the dissolved copper suppress the floatablilty of Cu and Mo minerals because oxide copper is not dissolved at the pH 8.
Author Response
we have revised based on your comments. please see the attachment, authors' reply and revised manuscript.
